# Rural Spatial Differentiation and Revitalization Approaches in China: A Case Study of Qingdao City

**DOI:** 10.3390/ijerph192416924

**Published:** 2022-12-16

**Authors:** Xiaohua Cheng, Difei Xu, Hui Sun, Meiyi Zheng, Jintao Li

**Affiliations:** 1School of Political Science and Public Administration, Shandong University, Qingdao 266237, China; 2Institute of Governance, Shandong University, Qingdao 266237, China; 3Institute of Quality of Life and Public Policy, Shandong University, Qingdao 266237, China

**Keywords:** rural revitalization, coupling coordination, three-dimensional location model, Qingdao

## Abstract

Rural revitalization, as a major strategy with the goal of realizing the overall development of strong agriculture industries, beautiful rural areas, and rich farmers, is an effective way of alleviating the loss of talent, land, capital, and other elements in rural areas and a possible cure for “rural diseases”. However, “rural diseases” faced by villages are very different, and thus exploring suitable strategies for rural revitalization is beneficial to the implementation of rural revitalization strategies and the promotion of urban–rural integration. Based on location theory, this paper constructs a point–axis–domain three-dimensional spatial location theory model that integrates market location, traffic location, and natural location and combines the coupling coordination model to comprehensively study the vitality and development directions of Qingdao’s rural areas. Results found that Qingdao’s high-level and medium–high-level coupling coordination areas are the main types of coupling coordination, accounting for 45.19% and 47.48%, respectively. Based on the development status of Qingdao, this study explores development directions for rural revitalization poles as well as high-level, medium–high-level, and medium-level coupling coordination areas and suggests the following: rural revitalization poles should play a demonstration role in rural revitalization in terms of industrial development, rural civilization, social governance, public service construction, etc.; high-level coupling coordination areas should focus on building modern hi-tech agriculture and rural marine tourism industries; medium–high-level coupling coordination areas should strengthen the building of satellite towns and promote industrial transformation and upgrading; medium-level coupling coordination areas should actively develop ecological environment conservation models and establish a characteristic mountainous eco-tourism industry. Thus, the findings provide important scientific reference for the implementation of rural revitalization.

## 1. Introduction

For a long time, the issue of “agriculture, rural areas and farmers” has attracted much attention as an important task of national development in China. A report from the 19th National Congress of the Communist Party of China (CPC) in 2017 proposed a rural revitalization strategy and prioritized the development of agriculture and rural areas. In 2018, the CPC Central Committee and the State Council’s release of the “*Strategic Plan for Rural Revitalization (2018–2022)*” indicated that rural revitalization and development had reached a stage of concrete implementation. In 2022, in the context of the global spread of the COVID-19 epidemic and the urgent need for economic recovery, the CPC Central Committee and the State Council’s “*Implementation Opinions on Promoting Key Work of All-round Rural Revitalization in 2022*” pointed out that the key tasks of rural development, rural construction, and rural governance, especially grain production and modern agricultural development, should be carried forward in an orderly manner. In this regard, accelerating the rural revitalization strategy provides a fundamental guarantee that the current problems of rural development and construction will be solved [1]. However, due to the variations of factors such as population, land, industry, location, resources, and environment, there are certain differences in the direction of rural development. Therefore, analyzing the spatial differentiation characteristics of rural areas and exploring the development directions and strategies of rural revitalization suitable for different types of location conditions have important implications for the targeted implementation.

The issue of rural development has once again attracted extensive attention from scholars after the release of the rural revitalization strategy. On the basis of studies examining the process of rural transformation and development and spatial patterns, researchers have been paying close attention to organization and institution building, talent development, capital investment, and infrastructure construction in rural areas [2,3]. Some researchers have conducted research on rural social governance and its supporting institutions through the formulation of rural transformation and development plans [4,5]. It is envisioned that exploring more suitable optimization strategies for rural transformation and development is possible by integrating multi-disciplinary qualitative analysis, such as political science, economics, management, and sociology integrations [6], and by combining quantitative case analysis [7]. The essence of rural revitalization is a systematic process of reorganization of rural system elements, spatial reconstruction, and functional improvement [8]. It is necessary to recognize the geographical and spatial differences and complex diversity of regional rural systems to scientifically and systematically summarize their evolutionary characteristics [9,10,11] while following the laws of rural development in different types of areas, which thus allows for the organization of spatial reconstructions and improvements in the function of rural regional systems [12]. Therefore, the rural spatial location, as an important factor affecting the development of rural industries and the survival of farmers, has received continued attention from scholars. Famous location theories that were initially developed by German economists, such as Thunnen’s agricultural location theory, Weber’s industrial location theory, and Chris Taylor’s central place theory [13,14,15], propose that traffic location and market location are important factors affecting the production, transportation, and sale of agricultural products. These theories provide important support for social and economic activities [16]. Researchers usually treat location as one of the main driving factors for urban and rural development and detect these driving mechanisms through spatial pattern and scale [17,18]. It is also suggested that, on the one hand, location can significantly affect the production, transportation, and sale of products, while on the other hand, human activities can greatly improve locations. Hence, under the combined effect of multiple factors, the income and overall quality of life of producers should be significantly improved.

After persistent research and exploration, scholars have put forward a number of new location theories, including the modern location theory (with scale and competitiveness as its core component) and the point–axis theory (focusing on the link between urban centers and traffic arteries) [19,20,21]. The point and axis locations are important factors affecting the urban–rural relationship. The urban center is the main distribution area of rural agricultural products and the supply center of public services, which has an important impact on rural industrial development and farmers’ lives. The main route connecting the urban center and the countryside is the traffic road. These convenient roads are conducive to accelerating the circulation of urban and rural elements. Rural agricultural products can quickly reach the urban distribution market and then connect with consumer groups outside the region. At the same time, the developed traffic road network is more conducive to attracting urban social capital that can be integrated into the countryside and which can provide sustainable impetus for the development of rural industries [22,23]. Overall, research findings guided by these location theories demonstrate that traffic roads, urban centers, and industrial scales can generate centripetal and centrifugal forces in urban and rural areas, thus affecting their scale of development and the direction of industrial development.

In Qingdao, an economically developed region on the east coast of China, issues such as urban–rural disparity and the unbalanced distribution of resources and facilities between urban and rural areas have become increasingly prominent alongside its rapid urbanization and industrialization [24]. On 30 December 2018, the Qingdao Municipal Party Committee and Government issued the “*Qingdao Rural Revitalization Strategic Plan (2018–2022)*”, and in August 2019 they issued the “*Qingdao Rural Revitalization Comprehensive Mission Scheme (2019–2022)*”. These documents further clarified the goals and tasks required for six tough missions related to the transformation and upgrading of rural industries, the diffusion of the “Laixi experience”, better ecological living conditions, enhanced talent aggregation, stronger cultural prosperity, and improved innovation in rural areas [25,26]. In the new era, Qingdao should accelerate the pace of urban–rural transformation, narrow the gap between urban and rural areas, promote integrated development through the rural revitalization strategy, and provide guarantees for the transition of old economic engines to new ones. Therefore, based on location theory, this paper (1) constructs a point–axis–domain three-dimensional spatial location theory model, (2) analyzes the spatial distribution advantages of Qingdao’s rural areas, (3) identifies the revitalization poles of rural transformation and development, and (4) suggests directions and development paths suitable for different areas in Qingdao. The findings should provide reference for Qingdao’s comprehensive rural revitalization missions and the integrated development of urban and rural areas.

## 2. Study Area and Research Methods

### 2.1. Area Overview

Qingdao is located in the southeast of Shandong Peninsula next to the Yellow Sea. It is an important central city along the coast of China. Currently, the city is affiliated with seven districts and three county-level cities. There are 6626 neighborhood committees and 5419 of them are village committees, accounting for 81.78% (Figure 1). Its total area is 11,293 km^2^ and the built-up area is 758.16 km^2^, with a population of 10.257 million residents and an urbanization rate of 77.17%. The city’s GDP is CNY 1.41 trillion, and the proportions of primary, secondary, and tertiary industries are 3.3%, 35.9%, and 60.8%, respectively. The annual grain sown area is 480,000 hm^2^ with a total grain output of 3.128 million tons, which is a 2.7% increase over the previous year. Currently, there are 1055 certified products that align with the “three products and one standard” principle(pollution-free agricultural products, green food, organic agricultural products, and geographical indications of agricultural products).

In 2021, Qingdao prioritized rural revitalization as one of the key strategic tasks of the city’s 15 comprehensive missions and has been actively exploring large-scale land management, village layout adjustment, land resource consolidation, beautiful countryside construction, rural complex construction, and the “five revitalizations” promotion mechanisms through organizational revitalization in rural areas. It aimed to promote agricultural modernization by increasing the scale of land management, commercializing organizations, modernizing technology, professionalizing service delivery, and marketizing business operations. In this regard, determining how to identify the shortcomings of Qingdao’s rural development so as to alleviate the bottlenecks associated with rural revitalization are among the outstanding issues yet to be solved.

### 2.2. Location Indicators and Data Processing

According to the theories of agricultural location, industrial location, and central place, the distance between a city center and major traffic arteries is an important factor affecting the production and sale of agricultural and industrial products, which also can be adopted to explain urban and rural disparity [27,28]. In addition, the natural environment plays an important role in agricultural production and rural development, e.g., wheat cultivation in northern China and rice cultivation in southern China. This implies that location elements affecting rural development are complex and diverse, and there are considerable differences between rural areas in different regions, especially in terms of the location of nature, traffic, and markets. By definition, natural location (NL) refers to the natural environment in which a village is located, which has an important impact on people’s living environment, crop planting structures, industrial development structures, etc. Traffic location (TL) refers to the relative position of major traffic arteries (such as highways and railways) around where a village is situated, which has an important impact on residents’ travel and product transportation. Market location (ML) refers to the distance between a village and its upper-level administrative unit, which has an important impact on the convenience of residents’ lives and product sales. We therefore chose elevation and slope as the natural location (NL), the distance to the municipal city center and county center as the economic (market) location (ML), and the distance to major railways and highways as the traffic location (TL) to jointly construct a location indicator system that measures the impact of rural development. It should be noted that the traffic location mentioned in this paper mainly consists of linear factors, without taking into account point elements such as railway stations or road intersections. Although this is also an important factor affecting rural development due to the wide distribution of road intersections, it is difficult to obtain relevant data. Furthermore, the distribution of railway stations is sporadic and only has a significant impact on certain villages. Therefore, this paper selected linear roads as the axis location elements according to previous research. In addition, this paper mainly selected some location factors as case studies and will consider more location factors for in-depth analysis in future research.

In this study, digital elevation model (DEM) data provided by the National Geomatics Center of China were used to obtain elevation and slope data for Qingdao City by means of image correction and slope analysis. With the aid of ArcGIS 10.2 software(Products released by American Esri in 2013), through Euclidean distance analysis, the distances from each village (neighborhood) committee to major railways and highways and to city center and county centers were calculated. In order to more scientifically analyze the types of rural areas in Qingdao and eliminate the interference caused by administrative boundaries, this study converted all location indicators into 500 m resolution raster data and used spatial adjustment to perform spatial matching and overlapping. After that, unified standardization and mean normalization were performed on the acquired data (Figure 2), thereby generating a three-dimensional location factors system of point–axis–domain (Table 1). Specifically, standardization refers to the global conversion of indicator data values from “0” to “1” through the calculation of their maximum–minimum value (min-max); mean normalization refers to the process of summing the values of NL, TL, and ML indicators separately and calculating their corresponding average values. According to the equal interval classification method in ArcGIS, the natural location, traffic location, and market location were then categorized as low level (0~0.2], relatively low level (0.2~0.4], medium level (0.4~0.6], relatively high level (0.6~0.8], and high level (0.8~1.0] to represent their spatial distribution characteristics.

### 2.3. Spatial Analysis Model

#### 2.3.1. Euclidean Distance Model

Euclidean distance refers to the “ordinary” distance between two points in a multidimensional space [29]. This study used the Euclidean distance model to calculate the shortest distance from each village (neighborhood) committee to the point and axis elements (city center, county center, main railway, and main highway) in the spatial location with the aid of the neighborhood analysis tool in ArcGIS. These distances only represent the relative distance between villages and elements and are used to describe the near far relationship between villages and different elements. We obtained the spatial distribution data through spatial distance interpolation analysis in ArcGIS. The formula is as follows:


(1)
$$D\left(x\right)=\sqrt{\left(x_{i}-x_{j}\right)\left(y_{i}-y_{j}\right)}$$


In the formula, point *i* is the village (neighborhood) committee and point *j* is a point in the city center, county center, main railway, or highway. *D*(*x*) is the shortest distance between point *i* and *j*; *x_i_* and *x_j_* are the latitudes of point *i* and *j*; and *y_i_* and *y_j_* are the longitudes of point *i* and point *j*.

#### 2.3.2. Coupling Coordination Model

The coupling coordination model was used to analyze the level of coordinated development. Coupling degree (*CD*) refers to the interaction between two or more systems to achieve the dynamic relationship of coordinated development, which can reflect the degree of interdependence between systems. Coordination degree (TD) refers to the degree of benign coupling in the coupling relationship, which can reflect the quality of coordination [30,31]. The coupling coordination degree model (*OD*) includes the calculation of coupling degree, coordination index, and coupling coordination degree. This study analyzed the relationship between the village (neighborhood) committees’ three-dimensional location systems according to coupling coordination degree values.


(2)
TD=α×ML+β×TL+γ×NL



(3)
CD=ML∗TL∗NL((ML+TL+NL)3)3



(4)
$$OD=\sqrt{TD*CD}$$


In the formula, TD, *CD*, and *OD* represent the coordination degree, coupling degree, and coupling coordination degree of the village (neighborhood) committees’ three-dimensional location; *α*, *β*, and *γ* indicate the weights of market location, traffic location, and natural location indicators.

In accordance with their three-dimensional location coupling coordination degrees, the village (neighborhood) committees can be divided into different development levels using an equal interval classification method. Based on the coupling coordination degrees of the market, traffic, and natural location, this study categorized the villages of Qingdao into five types: low-level coupling coordination areas (0~0.2], medium–low-level coupling coordination areas (0.2~0.4], medium-level coupling coordination areas (0.4~0.6], medium–high-level coupling coordination areas (0.6~0.8], and high-level coupling coordination areas (0.8~1.0]. The higher the coupling coordination degree value, the higher the level of division, which represents the given village (neighborhood) committee’s higher level three-dimensional location. Specifically, low-level coupling coordination means that the market, traffic, and natural locations are extremely uncoordinated and no obvious connection exists between them; medium–low-level coupling coordination indicates that there is a certain relationship between the market, traffic, and natural locations, though the degree of their mutual influence is relatively low; medium-level coupling coordination means that the relationship between the market, traffic, and natural locations is relatively clear, and there is certain interaction between them; medium–high-level coupling coordination means that the market, traffic, and natural locations have a relatively high-impact relationship; high-level coupling coordination means that the market, traffic, and natural locations are extremely coordinated, and they are jointly affecting the development of the given village (neighborhood) committee. The rural revitalization pole refers to a list of areas with a coupling coordination value greater than or equal to 0.95, indicating that the market, traffic, and natural location values for this type of rural area are all close to 1.0.

#### 2.3.3. Three-Dimensional Location Model

The point–axis–domain three-dimensional location theory model is comprised of three types of location elements: point, line, and area (Figure 3). According to the location theory and point–axis theory, rural revitalization and development are not determined by a single location factor, but are instead jointly influenced by natural, traffic, and market locations. These factors affect the production, processing, transportation, and sale of agricultural products in rural areas, as well as people’s daily lives and external communication. They are an important guarantee of sustainable development of agriculture and improvements in farmers’ living standards. To further measure the three-dimensional location status of each village (neighborhood) committee, a coupling coordination model was introduced to comprehensively analyze the coordination relationship between market, traffic, and natural locations so as to determine the location stability of village (neighborhood) committees. In the point–axis–domain location theory model, point *C* is the ideal location where the values of natural, traffic, and market locations are all 1 and the coupling coordination degree between them is 1, with the points on the *OC* connection also equal to 1. As the point moves far away from point *O* and comes close to point C, the comprehensive value of the point location becomes larger; the closer it is to *OC*, the greater the coupling coordination degree of that point is. For example, the comprehensive value of point B is higher than that of point *A*, but its coupling coordination degree is lower than point *A* (Figure 3). The three-dimensional location model shows that the higher the coupling coordination degree, the more obvious the rural market, transportation, and natural location advantages are and the higher the industrial development potential is, which is an important field for rural revitalization.

## 3. Results

### 3.1. Spatial Distribution Characteristics of Three-Dimensional Location Factors

According to the spatial distribution characteristics of three-dimensional location factors, the distribution characteristics of elevation and slope in the natural location were similar, with characteristic high values in the north and south edges and low values in the middle. In particular, the middle and northern part of Pingdu, Laoshan District, and southeast coastal areas of Jimo District were higher than other areas, with the highest values of elevation and slope being 309.79 m and 26.59°, respectively (Figure 4). The longest distance to the main highway in the traffic location was 54.07 km, which occurred in the northern part of Pingdu and the central coastal areas of Jimo District. The longest distance to the main railway was 89.18 km, which was mainly distributed in the southwest coastal areas of Huangdao District. The longest distance to the city center in the market location was 105.25 km, which occurred in the northern part of Pingdu and stretched radially to the northwest from Shinan District and Shibei District on the southeast coast. The distance to the county center was distributed in concentric circles around the county’s central areas and the highest value was 46.68 km, which was primarily distributed in the southwest coastal areas of Huangdao District and the eastern coastal areas of Jimo District. By and large, the market, traffic, and natural locations of Qingdao’s counties and districts exhibit obvious spatial differentiation characteristics, and the differences between regions are significant.

In order to eliminate the influence of unit dimensions from different location factors, a grid algebra calculator was employed to standardize (from 0 to 1) the three-dimensional location factors. Results indicate that the spatial distribution characteristics of the standardized factors are consistent with those of the original factors (Figure 5), which implies that the standardized values did not change the spatial distribution of location factors or their differences, and therefore can be used for in-depth analysis.

By averaging the standardized values of the three-dimensional location factors, the spatial distribution results of natural, traffic, and market locations were obtained. As shown in Figure 6, the spatial differentiation characteristics have obvious differences and layers. The proportion of villages with high-level natural locations was the largest, accounting for 81.77%, and there were no low-level areas. Most of the villages in Shinan District, Shibei District, Jimo District, and Jiaozhou City included high-level natural locations; villages with relatively low-level natural locations were mainly located in the northern part of Pingdu City and the eastern part of Laoshan District, accounting for 3.06%. Traffic location presents a spatial distribution structure with Laixi City–Jimo District–Chengyang District–Shibei District as the central axis, gradually decreasing to both sides; villages with high-level traffic locations were mainly located in Laixi City, Jimo District, Laoshan District, Jiaozhou City, and Chengyang District and accounted for 42.83% of traffic locations. Rural areas with low-level and relatively low-level traffic locations were mainly located in the northern part of Pingdu City and the eastern part of Jimo District, accounting for 1.92% and 6.84%, respectively. The proportions of villages with high-level and low-level market locations were relatively low, accounting for 2.38% and 1.13%, respectively. Villages with high-level market locations were mainly located in the border areas of Jimo District and Chengyang District, and villages with low-level market locations were mainly located in the northern part of Laixi City and the eastern part of Jimo District. Comparatively speaking, the proportion of villages with medium-level market locations was relatively large, accounting for 46.24%, and these locations were mainly found in Jimo District, Chengyang District, Jiaozhou District, Huangdao District, Shibei District, Shinan District, etc. Market locations manifested an obvious concentric structure, with Jimo District and Chengyang District as the center and gradually radiating locations present in surrounding areas.

Based on the above analysis, the spatial distribution characteristics of natural, traffic, and market locations in Qingdao’s rural areas exhibit certain regularity, and different villages have different location advantages. Therefore, it is of great value to explore rural revitalization strategies under the influence of different location advantages and to further promote the scientific and effective implementation of rural revitalization.

### 3.2. Coupling Coordination Characteristics of Three-Dimensional Location Factors

According to the distribution of the coupling coordination relationship between the nature, traffic, and market locations of Qingdao’s village (neighborhood) committees, there were no village (neighborhood) committees with low-level or medium–low-level coupling coordination in the study area, indicating a significant level of interaction between location factors. Results further indicated that the proportions of medium-level, medium–high-level, and high-level coupling coordination were 7.53%, 47.51%, and 45.23%, respectively. The highest proportion of villages (44.56%) was found in medium–high-level coupling coordination areas, while the highest proportion of neighborhood committees (15.97%) was found in high-level coupling coordination areas (Table 2). Moreover, the proportion of rural revitalization poles was 16.45%, among which villages and neighborhood committees accounted for 9.40% and 7.05%, respectively. Therefore, it can be inferred that Qingdao’s village (neighborhood) committees possess distinct location advantages and are higher level in terms of the three-dimensional location elements of the point–axis–domain model.

Spatially, the three-dimensional location coupling coordination of Qingdao village (residential) committees presents an obvious spatial hierarchical structure, which gradually expands from the center to the peripheries (Figure 7). Medium-level coupling coordination was mainly distributed in the northern areas of Pingdu City and the eastern coastal areas of Jimo District; high-level coupling coordination was mainly distributed in the central areas and southeastern coastal areas of Qingdao, including Shinan District, Shibei District, Chengyang District, Jiaozhou City, etc.; medium–high-level coupling coordination was mainly located in the northern, eastern, and southwestern coastal areas of Qingdao, including Pingdu City, Laixi City, Jimo District, and Huangdao District. In order to further clarify the development path of Qingdao’s rural revitalization mission, this study selected village (neighborhood) committees with a coupling coordination degree of three-dimensional location factors higher than 0.95 as the rural revitalization poles, which were mainly distributed in two concentrated areas. The first was village (neighborhood) committees in Jiaozhou City and Chengyang District (a proportion of 5.08% in total, of which village committees accounted for 3.84%). The second was village (neighborhood) committees located in Shibei District, Jimo District, Chengyang District, and other areas (a proportion of 11.15% in total, of which village committees accounted for 5.43%, and all of which were all located within Jimo District). The findings reveal that the point–axis–domain three-dimensional location of Qingdao’s village (neighborhood) committees presents a clear concentric structure, and the coupling coordination level gradually decreases from the southeast coast to the northwest areas.

## 4. Discussion

### 4.1. Differences in Research by Rural Location

The location factors of different villages are obviously different. How to accurately find the advantages of different rural locations and make full use of them to promote rural development is an important means of rural revitalization. Compared to existing location theories and models [32], which transform natural resource endowments and human environmental conditions (transportation and market factors) into location factors, this paper innovatively constructed a three-dimensional location model. With the help of the coupling coordination model, we analyzed the relationship between natural location, market location, and transportation location and divided villages with different location advantages based on their different coupling coordination degrees. The results provide an important reference for the development path of rural revitalization according to local conditions.

The primary innovation of this paper is that the point–axis–domain three-dimensional location model is proposed. Compared to the traditional point–axis location theory and central location theory [33,34], the three-dimensional model better explains the distribution characteristics of multiple factors that affect rural development. It is more scientific and reasonable to comprehensively analyze rural location endowment by coupling natural and human factors with single rural natural or human conditions [35,36]. Utilizing the relationship among objective NL, TL, and ML data should comprehensively reflect the impact of human activities and natural conditions on rural development. Additionally, the location factors selected in this study mainly consist of objective data with obvious differences, which excludes the impact of human subjective factors on rural endowment differences [37,38]. Secondly, the classification of rural types in this paper is based on the coupling and coordination of three-dimensional elements of point–axis–domain data, rather than a judgment of the type of rural development through comprehensive weighted summation of elements [39]. The coupling coordination analysis can not only retain the data characteristics of the original three aspects of elements while avoiding the impact of large variation in data on the overall evaluation results, but also examine the relationship between point–axis–domain three-dimensional elements through the coupling coordination analysis; this allows for clearer identification of the different types of rural location advantages and the proposal of an innovation path suitable for rural development. However, there are still some deficiencies in the selection of location data in this study. In particular, the NL elements could include climate factors, the TL elements could include more roads (such as provincial roads, county roads, etc.), and the ML elements could also consider towns and other surrounding cities. Due to limitations relating to data acquisition, we mainly chose six common and representative location elements to represent ML, TL, and NL in this study. In order to continuously supplement and improve the theoretical three-dimensional location model, we will strengthen the screening of more location element indicators to make the theoretical rural three-dimensional location model more substantial.

According to the results, the point–axis–domain three-dimensional location factors exert different effects on urban and rural areas. The point element is mainly the market location, which is the main distribution center of rural agricultural products and the location where education, medical care, and other important public services are provided. This location offers an important consumer market for the development of rural industries. The axis element is the main traffic location and an important bridge connecting villages and towns. Convenient traffic trunk lines can transport rural agricultural products to different urban consumer markets. At the same time, they can also provide sufficient tourist resources for the development of rural tourism industries and attract foreign capital to drive rural development. The domain elements are the main natural locations and the basic guarantee of rural development. The flat areas have sufficient arable land resources, which are suitable for the development of large-scale modern agriculture. The steep mountainous areas have rich forest land and grassland resources. With the help of local natural scenery and the cultural landscape, rural leisure tourism and a sightseeing industry can be developed, which is an important factor when seeking to attract foreign tourists. Thus, the three-dimensional location elements provide an important guarantee of rural development, promote the flow between urban and rural elements, and are the key factors of strengthening urban–rural integration.

In recent years, with the proposal of China’s rural revitalization strategy, there have been more and more studies on rural development. Unlike foreign rural studies, Chinese scholars have paid more attention to the topics of rural population, industry, culture, organization, and ecology, mainly in response to China’s strategic needs [40,41]. The research on rural revitalization methods in this paper is based on the spatial analysis of location elements, with the aim of exploring the relationship between different location endowments in rural areas and industrial revitalization. Compared to research on rural education, medical care, health, ecological environments, and other specific issues in developed countries [42,43], the research results of this paper are mainly useful for the formulation of strategic planning that can be used for rural revitalization and the selection of industrial development models. However, as this paper only chooses Qingdao as an example for testing the three-dimensional location model, it is important to note that there are a large number of villages in other regions that are different to those in Qingdao. Therefore, the main purpose of this paper is to introduce a research method for exploring rural revitalization approaches with the help of three-dimensional location elements and provide a new perspective and research ideas for empirical research on rural development.

### 4.2. Rural Revitalization Approaches in Different Coupling Coordination Areas

Spatial location is an important factor affecting the development of rural industries and farmers’ livelihoods. By means of spatial aggregation and coupling coordination analysis, this study has identified the revitalization poles of Qingdao’s rural areas and the high-level, medium–high-level, and medium-level coupling coordination areas of three-dimensional location factors. In order to better promote the implementation of rural revitalization strategies and build a Qilu model for rural revitalization in Qingdao, this research based on the corresponding relationship between the coupling coordination areas of location factors and different location elements (Figure 8) proposes revitalization and development paths that are suitable for different types of rural villages in Qingdao.

Rural revitalization poles are mainly located in Qingdao’s central urban areas, such as Jimo District, Shibei District, Chengyang District, Jiaozhou City, etc. The natural, market, and traffic locations of this type of village committee have distinct advantages. Their standardized value exceeds 0.9, which means that they can be regarded as the core areas of industrial development and represent the highest level of social and economic development in Qingdao. Meanwhile, the analysis of spatial distribution characteristics reveals that the village committees of rural revitalization poles in Qingdao are mainly located in Jimo District and Jiaozhou City. Therefore, villages in Jimo District and Jiaozhou City should be treated as pilot demonstration zones for rural revitalization industries (with a focus on rural civilization, social governance, public service provision, etc.) to accelerate the equalization of basic public service provision in rural areas and the integrated development of urban and rural areas. These locations can also take advantage of the latest science and technology improvements to develop smart agriculture that focuses on promoting the high-tech gene breeding industry and smart agriculture facilities so that they can become the core areas of rural industrial technology and important markets for agricultural products in Qingdao. Given their pioneering role in promoting the integrated development of primary, secondary, and tertiary industries in rural areas, rural revitalization poles can lead the way in development of the five revitalization fields of industry, talents, culture, ecology, and organization.

The high-level coupling coordination areas, accounting for 45.19% of the total, are located in Qingdao’s urban center and include Huangdao District, Laoshan District, Chengyang District, etc. For this type of area, more than 80% of the village (neighborhood) committees had a standardized distance to the main highway, distance to the main railway, and slope and elevation higher than 0.8, while only 30% of the village (neighborhood) committees had a standardized distance to the city center and county center higher than 0.8, which implies that high-level coupling coordination areas have a greater advantage in terms of traffic locations and natural locations. These village (neighborhood) committees are mainly located in the core areas of urban economic development and land expansion, where large-scale agricultural development is restricted due to the lack of land resources. Therefore, the rural revitalization strategy should focus on controlling land expansion, limiting population size, and playing a leading role in rural economic development in medium-level and medium–high-level coupling coordination areas by constructing high-tech agriculture and industry zones. At the same time, by taking advantage of the rich marine resources, these villages could actively develop characteristic industries, such as marine tourism or leisure homestay industries.

The medium–high-level coupling coordination areas (accounting for 47.48% of the total) possess the largest number of village (neighborhood) committees, the majority of which are located in the areas of Pingdu City, Laixi City, and Jimo District. This type of areas’ standardized slope and elevation values are relatively high, with more than 60% of village (neighborhood) committees higher than 0.8. Meanwhile, less than 20% of village (neighborhood) committees have a standardized distance to the main highway and railway higher than 0.8, and less than 30% have a standardized distance to the city center and county center higher than 0.6. This implies that these village (neighborhood) committees are favorable in terms of natural locations but less favorable in terms of traffic and market locations due to their considerable distance to the city and county center. Although arable land resources in these areas are abundant alongside flat rural terrain, the level of traffic and quality of economic locations is low, which hinders the development of their rural industrial economy. This type of area is located in major agricultural planting areas and is important for Qingdao’s industrial economic development. These areas should take advantage of the latest technologies and mature markets that rural revitalization poles and high-level coupling coordination areas provide so they can develop large-scale agriculture to provide sufficient agricultural products for Qingdao and its surrounding areas, especially in fields with Qingdao characteristics such as grain production and vegetable and fruit planting. Moreover, under the guidance of relevant policies, these villages should strengthen capital investment, improve the level of public services in the region by building urban centers with high-standard service facilities, and accelerate the construction of efficient logistics markets within the region.

The medium-level coupling coordination areas are distant from the urban center of Qingdao, mainly located in the north of Pingdu City and the east of Jimo District, and the proportion of village (neighborhood) committees in these areas is 7.33%. Their market and traffic locations are poor, with more than 90% of village (neighborhood) committees having a standardized distance to the main highway, main railway, city center, and county center lower than 0.6. In contrast, the level of their natural locations is comparatively high, with more than 60% of village (neighborhood) committees having a standardized value of elevation and slope higher than 0.6. This implies that these village (neighborhood) committee’s development levels in terms of rural population, land, and industry are considerably lower than those of other types of areas. Their industry is dominated by traditional agricultural production, and the majority of the population is working away from their hometown. Moreover, the traffic, health care, and education infrastructure in these areas is not capable of attracting farmers from neighboring locations due to the distance to the city center. Nevertheless, these areas are rich in mountain and forest tourism resources and could be reserved as ecological environment conservation areas to establish a characteristic mountainous eco-tourism industry.

## 5. Conclusions

Rural revitalization is a major strategy for promoting social and economic development in rural areas and improving farmers’ living standards. Different types of villages possess different advantages in terms of their location and resource endowments. Therefore, to design rural revitalization implementation strategies tailored to local conditions, it is necessary to investigate and clarify the advantages and endowments of different types of villages. This study took Qingdao, Shandong as an example and constructed a point–axis–domain three-dimensional spatial location theory model that integrated market, traffic, and natural locations. Additionally, based on the coupling coordination analysis model, this paper categorized villages in Qingdao into medium, medium–high, and high-level coupling coordination areas and clarified their location advantages, rural revitalization industries, and potential functions for different villages. Meanwhile, in order to better promote the formulation of rural revitalization strategies and the implementation of development approaches under the guidance of the three-dimensional location theory model, additional research focusing on micro-level cases of rural revitalization is needed.

## Figures and Tables

**Figure 1 ijerph-19-16924-f001:**
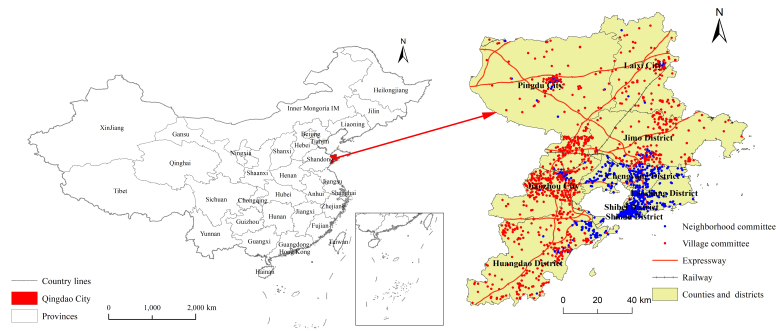
Distribution of the study area.

**Figure 2 ijerph-19-16924-f002:**
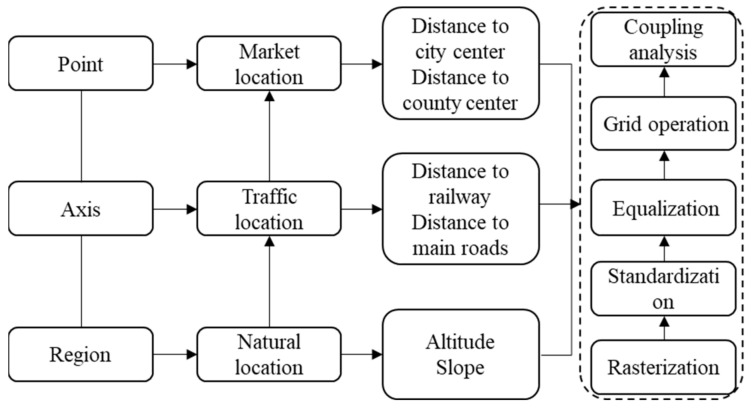
Three-dimensional location data processing process.

**Figure 3 ijerph-19-16924-f003:**
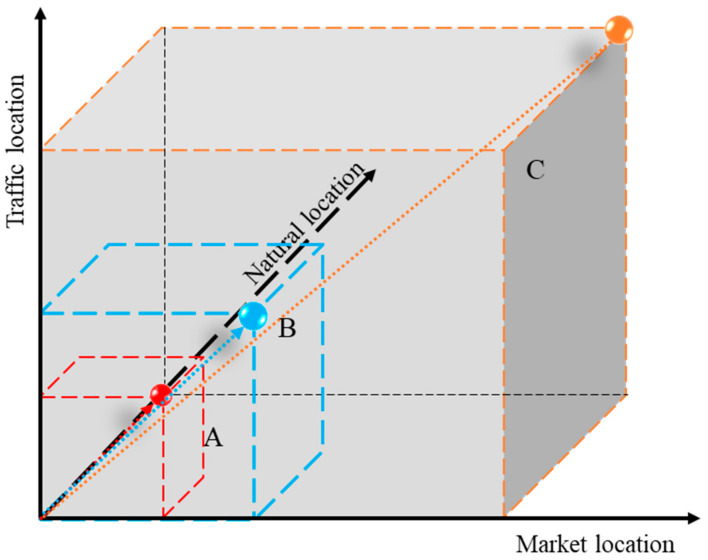
Point–axis–domain three-dimensional location model.

**Figure 4 ijerph-19-16924-f004:**
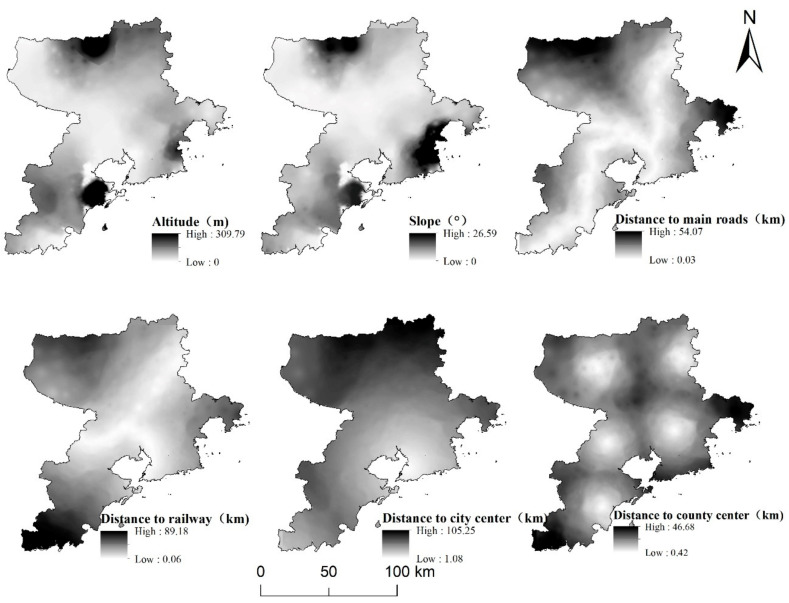
Spatial distribution of three-dimensional location factors.

**Figure 5 ijerph-19-16924-f005:**
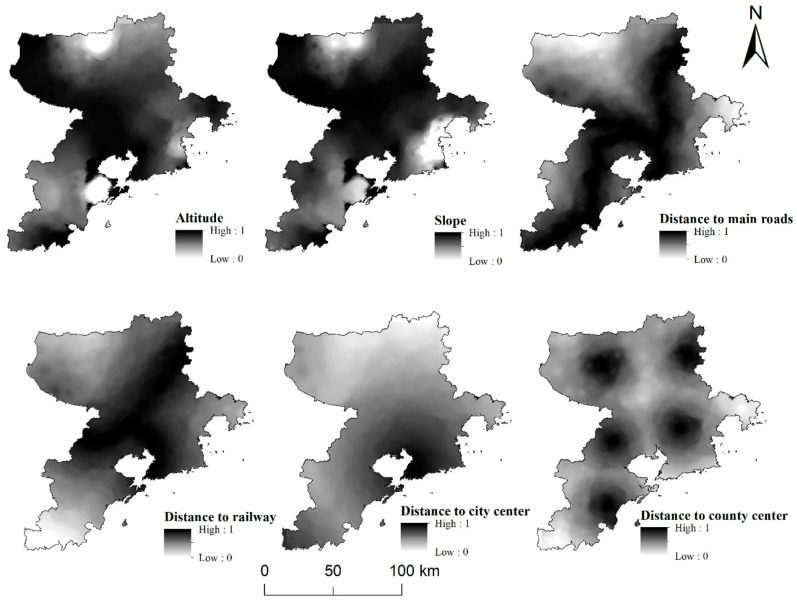
Standardized spatial distribution of three-dimensional location factors.

**Figure 6 ijerph-19-16924-f006:**
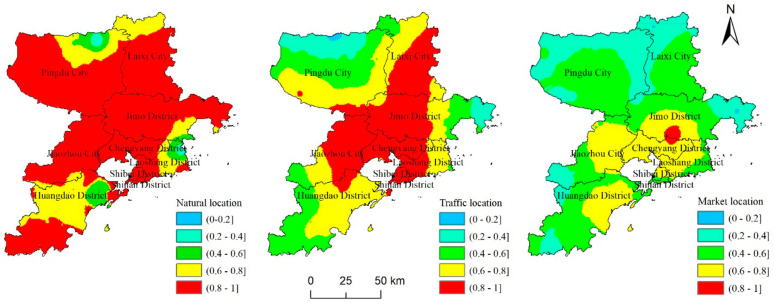
Spatial distribution of point–axis–domain three-dimensional locations.

**Figure 7 ijerph-19-16924-f007:**
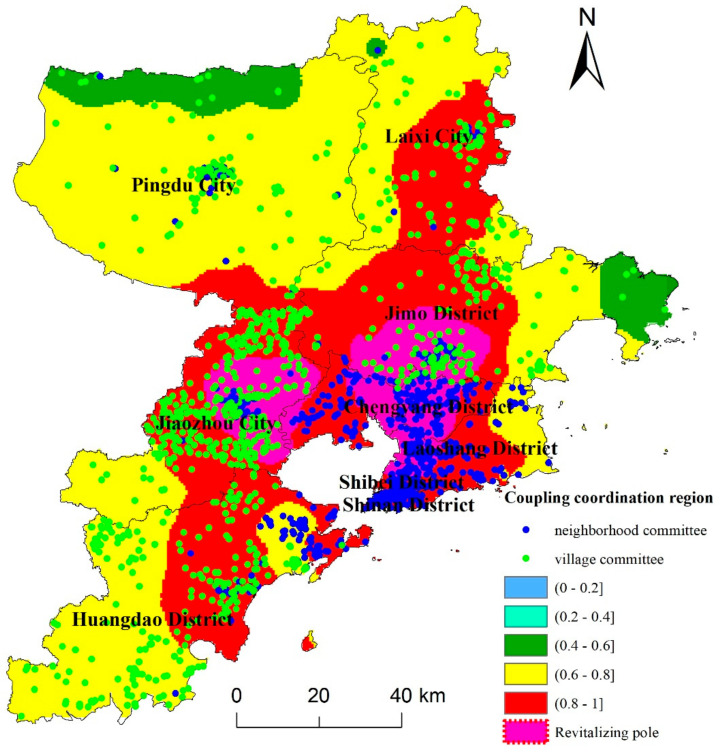
Distribution of coupling coordination zones for rural revitalization in Qingdao.

**Figure 8 ijerph-19-16924-f008:**
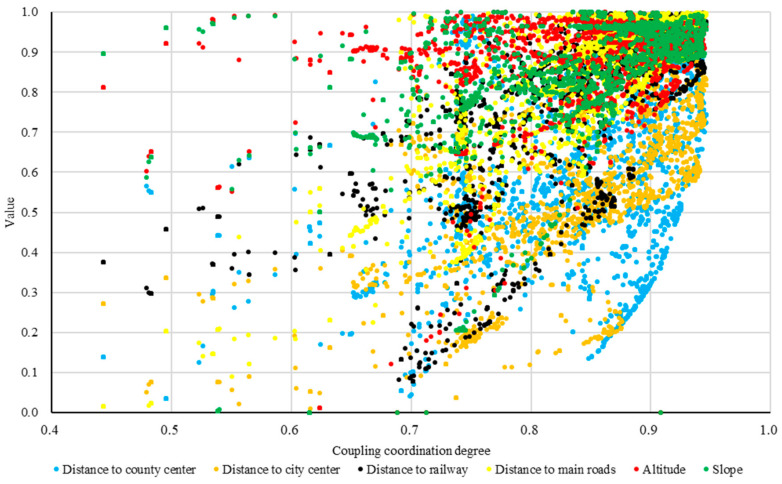
Correspondence between coupling coordination zones and location elements of rural revitalization.

**Table 1 ijerph-19-16924-t001:** Point–axis–domain three-dimensional location elements.

Target Layer	Factor Layer	Element Layer
Point	Market location (ML)	Distance to city center (*x*_5_)
		Distance to county center (*x*_6_)
Axis	Traffic location (TL)	Distance to major railways (*x*_3_)
		Distance to major highways center (*x*_4_)
Domain	Natural location (NL)	Elevation (*x*_1_)
		Slope (*x*_2_)

**Table 2 ijerph-19-16924-t002:** Three-dimensional location coupling and coordination distribution of village (neighborhood) committees.

	Village Committee	Neighborhood Committee	Total
Medium-level coupling coordination	7.23	0.1	7.33
Medium–high-level coupling coordination	44.56	2.92	47.48
High-level coupling coordination	29.22	15.97	45.19
Total	81.01	18.99	100

## Data Availability

Data will be available on request.

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
