# Peer review of "Rural Spatial Differentiation and Revitalization Approaches in China: A Case Study of Qingdao City"

_ijerph, 2022, doi:10.3390/ijerph192416924_

Round 1

Reviewer 1 Report

Location theory plays an important role in regional development, and location is also crucial in urban and rural development. This paper constructed a point-axis-domain three-dimensional spatial location theory model that integrates market location, traffic location and natural location, and combined the coupling coordination model to comprehensively study the vitality and development directions of Qingdaos rural areas. This can enable us to quickly identify which villages or communities have better location advantages, and can also implement urban and rural development policies according to the location. Therefore, this paper is innovative and has certain theoretical and practical value. However, as far as the current paper is concerned, there are still some contents that need to be carefully revised. They are as follows:

(1) In the summary part, it is suggested to summarize the results of the study rather than specific place names. It is suggested to summarize the results based on the characteristics of such areas.

(2) In the introduction, it is suggested to supplement the discussion on the relationship between location and urban and rural development.

(3) It is suggested to explain why the distance to the railway station or the intersection of the main road is not needed.

(4) It is suggested to add discussion on how the location affects the overall urban and rural integration in the discussion section.

(5) It is suggested to check the syntax of the full text, such as when to use the present tense and when to use the past tense.

Author Response

Location theory plays an important role in regional development, and location is also crucial in urban and rural development. This paper constructed a point-axis-domain three-dimensional spatial location theory model that integrates market location, traffic location and natural location, and combined the coupling coordination model to comprehensively study the vitality and development directions of Qingdao’s rural areas. This can enable us to quickly identify which villages or communities have better location advantages, and can also implement urban and rural development policies according to the location. Therefore, this paper is innovative and has certain theoretical and practical value. However, as far as the current paper is concerned, there are still some contents that need to be carefully revised. They are as follows:

(1) In the summary part, it is suggested to summarize the results of the study rather than specific place names. It is suggested to summarize the results based on the characteristics of such areas.

According to the reviewer’s suggestion, in the summary, we summarized the results based on the characteristics of all coupling coordination areas.

“rural revitalization poles should play a demonstration role of rural revitalization in terms of industrial development, rural civilization, social governance, public service construction, etc; The high-level coupling coordination areas should focus on building modern hi-tech agriculture and rural marine tourism industry; The medium-high level coupling coordination areas should strengthen the building of satellite towns and promote industrial transformation and upgrading; The medium-level coupling coordination areas should actively develop the model of ecological environment conservation and establish a characteristic mountainous eco-tourism industry. Thus, the findings provide important scientific reference for the implementation of rural revitalization.”

(2) In the introduction, it is suggested to supplement the discussion on the relationship between location and urban and rural development.

According to the reviewer’s suggestion, we added the discussion on the relationship between location and urban and rural development in the introduction.

“The point and axis locations are important factors affecting the urban-rural relationship. The urban center is the main distribution area of rural agricultural products and the sup-ply center of public services, which has an important impact on rural industrial development and farmers' lives. The main route connecting the urban center and the country-side is the traffic road. The convenient road is conducive to accelerating the circulation of urban and rural elements. Rural agricultural products can quickly reach the urban distribution market, and then connect with the consumer groups outside the region; At the same time, the developed traffic road network is more conducive to attracting urban social capital to integrate into the countryside and providing sustainable impetus for the development of rural industries [22-23].”

(3) It is suggested to explain why the distance to the railway station or the intersection of the main road is not needed.

According to the reviewer’s suggestion, we added the reason why we not considered the distance to the railway station or the intersection of the main road.

“It should be noted that the traffic location mentioned in this paper is mainly linear factors, without taking into account the point elements such as railway stations or road intersections. Although this is also an important factor affecting rural development due to the wide distribution of road intersections, it is difficult to obtain relevant data. Furthermore, the distribution of railway stations is sporadic and only has a significant impact on certain villages. Therefore, this paper selected linear roads as the axis location elements according to former research. In addition, this paper mainly selected some location factors as case studies, and will consider more location factors for in-depth analysis in future re-search.”

(4) It is suggested to add discussion on how the location affects the overall urban and rural integration in the discussion section.

According to the reviewer’s suggestion, we added discussion on how the location affects the overall urban and rural integration in the discussion section.

“According to the results, the point-axis-domain three-dimensional location factors exert different effects on urban and rural areas. The point element is mainly the market location, which is the main distribution center of rural agricultural products and the supply place of education, medical care and other important public services. It provides an important consumer market for the development of rural industries. Axis element is the main traffic location and an important bridge connecting villages and towns. Convenient traffic trunk lines can transport rural agricultural products to different urban consumer markets. At the same time, it can also provide sufficient tourist resources for the development of rural tourism industry and attract foreign capital to drive rural development. Domain elements are the main natural locations and the basic guarantee for rural development. The flat areas have sufficient arable land resources, which are suitable for the development of large-scale modern agriculture. The steep mountainous areas have rich forest land and grassland resources. With the help of local natural scenery and cultural landscape, rural leisure tourism and sightseeing industry can be developed, which is an important factor to attract foreign tourists. Thus, the three-dimensional location elements provide an important guarantee for rural development, promote the flow between urban and rural elements, and are the key factors to strengthen urban-rural integration.”

(5) It is suggested to check the syntax of the full text, such as when to use the present tense and when to use the past tense.

According to the reviewer’s suggestion, we checked the syntax of the full text.

Reviewer 2 Report

The paper is on an important issue and presents sufficient new information for publication. However, results discussion may be improved and methods may be explained in a plain language for the general readers of the journal. 

Author Response

The paper is on an important issue and presents sufficient new information for publication. However, results discussion may be improved and methods may be explained in a plain language for the general readers of the journal. 

 According to the reviewer’s suggestions, we supplemented and improved the results discussion and research methods.

“These distances only represent the relative length between villages and elements, and are used to describe the near far relationship between villages and different elements. We obtained the spatial distribution data through the spatial distance interpolation analysis in ArcGIS.”

“The three-dimensional location model shows that the higher the coupling coordination degree, the more obvious the rural market, transportation and natural location ad-vantages are, and the higher the industrial development potential is, which is an important field for rural revitalization.”

“According to the results, the point-axis-domain three-dimensional location factors exert different effects on urban and rural areas. The point element is mainly the market location, which is the main distribution center of rural agricultural products and the supply place of education, medical care and other important public services. It provides an important consumer market for the development of rural industries. Axis element is the main traffic location and an important bridge connecting villages and towns. Convenient traffic trunk lines can transport rural agricultural products to different urban consumer markets. At the same time, it can also provide sufficient tourist resources for the development of rural tourism industry and attract foreign capital to drive rural development. Domain elements are the main natural locations and the basic guarantee for rural development. The flat areas have sufficient arable land resources, which are suitable for the development of large-scale modern agriculture. The steep mountainous areas have rich forest land and grassland resources. With the help of local natural scenery and cultural landscape, rural leisure tourism and sightseeing industry can be developed, which is an important factor to attract foreign tourists. Thus, the three-dimensional location elements provide an important guarantee for rural development, promote the flow between urban and rural elements, and are the key factors to strengthen urban-rural integration.”

Reviewer 3 Report

The manuscript focuses rural spatial differentiation and revitalization approaches referring to the exemplary case study of Qingdao City. Based on the location theory the authors highlight the development of a point-axis domain three-dimensional spatial location theory model that integrates market location, traffic location and natural location. Overall abstract, introduction, study methodology framework and result sections are written well.

Though the paper refers to a single-spot case study, referring to the introduction as well as conclusion sections, the authors should clearly integrate more international research study findings, to highlight pros and cons of the presented model approach more accurately in view of previous research.

Author Response

The manuscript focuses rural spatial differentiation and revitalization approaches referring to the exemplary case study of Qingdao City. Based on the location theory the authors highlight the development of a point-axis domain three-dimensional spatial location theory model that integrates market location, traffic location and natural location. Overall abstract, introduction, study methodology framework and result sections are written well.

Though the paper refers to a single-spot case study, referring to the introduction as well as conclusion sections, the authors should clearly integrate more international research study findings, to highlight pros and cons of the presented model approach more accurately in view of previous research.

According to the reviewer’s suggestions, we added the pros and cons of the presented model approach more accurately in view of previous research in discussions.

“The primary innovation of this paper is that the point-axis-domain three-dimensional location model is proposed. Compared with the traditional point-axis location theory and central location theory [33,34], the three-dimensional model better explains the distribution characteristics of multiple factors that affect rural development. It is more scientific and reasonable to comprehensively analyze the rural location endowment by coupling the natural and human factors with the single rural natural or human conditions [35,36]. Utilizing the relationship among objective data of NL, TL and ML, it should comprehensively reflect the impact of human activities and natural conditions on rural development. And the location factors selected in the study are mainly objective data with obvious differences, excluding the impact of human subjective factors on rural endowment differences [37,38]. Secondly, the classification of rural types in this paper is based on the coupling and coordination of three-dimensional elements of point-axis-domain, rather than judging the type of rural development through comprehensive weighted summation of elements [39]. The coupling coordination analysis can not only retain the data characteristics of the original three aspects of elements, avoid the impact of large variation data on the overall evaluation results, but also examine the relationship between point-axis-domain three-dimensional elements through the coupling coordination analysis, so as to more clearly identify the different types of rural location advantages, and then propose an innovation path suitable for rural development. How-ever, there are still some deficiencies in the selection of location data in this study. In particular, the NL elements can add climate factors, the TL elements can add more roads such as provincial roads, county roads, etc., and the ML elements can also consider towns and other surrounding cities. Due to the limitations of data acquisition, we mainly chose six common and representative location elements to represent ML, TL and NL respectively in this study. In order to continuously supplement and improve the three-dimensional location theoretical model, we will strengthen the screening of more location element indicators to make the rural three-dimensional location theoretical model more substantial.”

Round 2

Reviewer 3 Report

The authors optimized beforementioned critical issues and the mansucript should thus be ready for publication right now.